# "Being a man is like being put in a box": A qualitative study of adolescent boys' and young men's understanding and experiences of mental health in an urban community in South Africa

Christopher Barkley[1,2☺*], Sandile Mnculwane[3‡], Katherine G. Merrill[4‡], Zuhayr Kafaar[2‡]

**1** Grassroot Soccer, Inc., Cape Town, South Africa, **2** Department of Psychology, Stellenbosch University, Stellenbosch, South Africa, **3** Independent Educator and Researcher, Alexandra, South Africa, **4** Center for Dissemination and Implementation Science, University of Illinois Chicago, Chicago, Illinois, United States of America

☺ These authors contributed equally to this work.
‡ SM, KGM and ZK also contributed equally to this work.
\* cbarkley@grassrootsoccer.org

## Abstract

Adolescent boys and young men (ABYM) in South Africa face a high burden of unmet mental health needs but are often overlooked in research and practice. Economic and racial inequalities, masculine norms, and limited access to targeted mental health promotion services may hinder their ability to understand psychological distress, seek support, and engage with psychosocial services. This qualitative study explored how ABYM in Alexandra, South Africa, perceive and experience mental health, to inform future interventions. Semi-structured interviews were conducted with 24 participants, including 12 adolescent boys (ages 15–19) and 12 male youth mentors and staff from a local adolescent health organization. Interviews were analyzed using reflexive thematic analysis. Participants were often unfamiliar with the term *mental health* but described distress through everyday language grounded in social and emotional experience. Their conceptualizations of mental health were shaped by family and community environments, gendered expectations, and a mix of psychological and supernatural explanations. Rigid masculine norms associated mental health challenges with weakness, discouraged emotional expression and help-seeking, and placed early and significant pressure on boys to succeed in school or sports to make money and fulfil the male provider role. Despite these pressures, many participants expressed personal views that challenged dominant norms, for example, valuing emotional expression and open conversations with trusted adults or peers about mental health. However, stigma and a lack of youth- and male-friendly services remained significant barriers to accessing formal support. Our findings highlight the need for gender-sensitive, culturally grounded mental health programming for ABYM. Interventions should involve youth in design and delivery, build on familiar coping

**Data availability statement:** The qualitative data underlying this study are not publicly available due to ethical restrictions. The data consist of in-depth interview transcripts with adolescent boys (including minors) and young adult men from a small, identifiable community and include sensitive information related to mental health and personal experiences. Despite de-identification, complete anonymization cannot be guaranteed. The data are stored securely by the research team in accordance with institutional data protection and ethics requirements. Data are available upon reasonable request, subject to confidentiality safeguards. Requests for access should be directed to the Human Research Ethics Committee at Stellenbosch University (HREC Reference No: S23/08/197) via the institutional Research Data Management office ([rdm@sun.ac.za](mailto:rdm@sun.ac.za); (https://www.su.ac.za/en/faculties/medicine/research/ethics), which serves as a non-author institutional point of contact to ensure long-term data availability and oversight.

**Funding:** This work was supported by the Oak Foundation [Grant Number: OFIL-23-092] as part of a broader grant to Grassroot Soccer for the project "More than a game: Transforming Gender Norms Amongst Adolescent Boys." The funding supported the research activities reported in this manuscript. The funders had no role in study design, data collection and analysis, decision to publish, or preparation of the manuscript. https://oakfnd.org.

**Competing interests:** The authors have declared that no competing interests exist.

strategies, normalize emotional expression among boys, and promote the attributes of good mental health. Embedding support within community or recreational settings may reduce stigma, improve engagement, and strengthen adolescent boys' mental health and well-being in low-resource urban contexts.

## Introduction

Adolescence (10–19 years) is a time of profound physical, psychological, and social changes that provide unique opportunities to promote well-being and prevent disease [1]. Adolescence is when many lifelong health-related behaviors and habits are formed. Young people take greater responsibility for their health [2]. Adolescence and young adulthood are particularly critical life stages for mental health promotion and for the prevention of mental health disorders, given that 35% of mental health disorders onset by the age of 14 and 75% before the age of 25 years [3,4].

Globally, mental health and substance use conditions are the leading causes of disability-adjusted life years (DALYs) for adolescents aged 10–24 years for both sexes, with an estimated 1 in 4 adolescents suffering from mental health conditions [5,6]. A 2021 systematic review based on 37 studies from 16 countries in sub-Saharan Africa (SSA) published from 2008- 2020 reported a higher prevalence of mental health conditions among adolescents (10–19 years) than global averages and other middle-income countries [7]. Nationally representative surveys on the prevalence of mental health disorders in South Africa are rare, and the data on the prevalence of child and adolescent mental health conditions in South Africa are limited [8,9]. However, a 2022 study using symptom scales found that more than a quarter of South Africans reported moderate to severe symptoms of probable depression, ranging from 14.7% to 38.8% [10].

Gender identity and masculinity are important yet often underexplored determinants of adolescent mental health. Adolescents face gendered differences in terms of stressors, coping mechanisms, and expressions of psychological distress [11], with gender differences in mental disorders emerging during this developmental stage [12]. Masculine norms are social rules and expectations linked to manhood and shape boys' and men's health behaviors [13]. While biological factors contribute to sex-specific health outcomes, the greater burden of morbidity and mortality among men is largely attributed to health practices shaped by social and cultural influences [14].

Conformity to traditional masculine norms, such as risk-taking, emotional suppression, and self-reliance, has been shown to affect mental health and negatively deter help-seeking [15]. These norms are internalized early in life, limiting boys' capacity to develop healthy coping strategies for stress, trauma, or adversity [16]. Moreover, while gender differences in prevalence, symptomatology, and risk factors are well established [12], adolescent boys and young men (ABYM) are often overlooked in mental health policies and interventions [17].

In South Africa, race, class, and the legacies of colonialism and apartheid shape both mental health and masculinity [18–20]. Factors such as intergenerational

trauma, persistent socio-economic stressors, and unequal access to mental health care contribute to high levels of psychological distress in poor, urban, Black communities such as Alexandra [21,22]. Hegemonic masculinity has often been described as centring the man as an economic provider within a heterosexual household [23]. However, in a context of historical and ongoing economic and political dispossession of Black men, where they are structurally excluded from fully achieving this ideal, masculine norms emphasising physical strength, dominance, emotional suppression, and toughness have often developed [24–26]. Mental health experiences are deeply implicated in these processes, both shaping and being shaped by how boys and young men understand and perform masculinity within their specific social and historical context.

Help-seeking, defined by the American Psychological Association as seeking assistance through formal or informal means, is a critical component of mental health care, as untreated mental health problems are associated with poorer outcomes [27,28]. Yet, men are consistently less likely than women to access health services, including those for psychological distress [13]. Masculine ideals of stoicism and self-reliance can conflict with help-seeking behaviors, particularly among young men, who have some of the lowest rates of professional mental health service use [29,30]. In South Africa, where only 25% of those needing care receive it [31], recent efforts have integrated mental health services into primary care [32–34]. However, boys and men face unique barriers to accessing and engaging with these services, including difficulties initiating help-seeking, challenges with establishing therapeutic relationships, and lower rates of service retention [35,36].

Given these gaps, promoting mental health proactively before clinical intervention is essential. Understanding how adolescents conceptualize mental health is central to designing effective support strategies. Studies have shown that adolescents often use nonclinical language to describe distress [37–39] and that gender strongly shapes how mental health is understood and expressed [40,41]. However, few studies in South Africa have explored how ABYM perceive and talk about mental health. Instead, research has tended to focus on the relationship between masculine norms and behaviors, such as violence or sexual risk-taking [26,42]. Despite well-documented gender differences in mental health and help-seeking, little is known about how ABYM perceive and communicate about their mental health, particularly in low-resource settings such as Alexandra, South Africa. This study aimed to address these gaps in the literature by exploring ABYM's understanding of and experiences with mental health and help-seeking. Specifically, this study aimed to explore the following:

1. Perceptions of mental health and the life experiences that shape it.

2. Masculine norms and personal beliefs, and their influence on perceptions of mental health.

3. Coping strategies used in response to mental health challenges, and beliefs that enable or limit help-seeking.

## Methods

### Research design

This study employed a qualitative, exploratory design to allow for an in-depth exploration of ABYM's lived experiences, beliefs, and language [43,44] surrounding mental health and help-seeking in Alexandra, South Africa. Given the limited existing research on this topic in the local context, an exploratory approach was appropriate to generate context-specific insights into these topics. To enhance transparency and completeness of reporting, the study methods and results were mapped to the Consolidated Criteria for Reporting Qualitative Research (COREQ) checklist [45], with the completed checklist provided as Supporting Information (Appendix A).

### Study setting and participants

The participants were recruited from Grassroot Soccer (GRS) programs in Alexandra ("Alex"). GRS is a global adolescent health organization that has been working with its local GRS affiliate in South Africa for 20 years. The GRS affiliate has

delivered health programs to over 350,000 youth in South Africa and has operated in Alexandra since 2009. Alexandra is located within the city of Johannesburg metropolitan area, with an estimated population of ±750,000. Alexandra is a youthful and vibrant community, but it falls within the poorest income quintile in the city of Johannesburg [46], and faces challenges with overcrowded housing, high unemployment rates, limited access to socioeconomic opportunities, and poor service delivery [47].

The study sample included 24 participants, including 12 adolescent boys and 12 GRS youth mentors and male GRS staff. Adolescent participants were eligible if they self-identified as male, were between 15 and 19 years old, and resided in Alexandra. The key informants, GRS youth mentors and staff, were included based on their unique ability to work directly with adolescent boys in the community. These individuals are trained community-based educators, facilitators, and coordinators who deliver GRS programs across Alexandra. All youth mentors and staff receive training in social and emotional learning, mental health, and psychosocial support. They also undergo background checks, complete child safeguarding training, and sign codes of conduct. As slightly older males from the same community, they offer valuable insights into mental health needs, challenges, and support systems relevant to adolescent boys. The inclusion criteria for youth mentors and staff were that they identified as male, were 18 or older, and had worked with the GRS for at least three months.

## Sampling and recruitment

A purposive sampling strategy was used to recruit adolescent boys who met the eligibility criteria of being 15–19 years old, residing in Alexandra, and having participated in GRS programming. Purposive sampling was chosen because the study sought detailed and contextually relevant accounts from participants with direct experience of the social and cultural environment under investigation [48]. Key informant interviews with GRS youth mentors and staff were included to provide additional perspectives to contextualize findings from the adolescent interviews.

Recruitment took place from February 1 to February 25, 2024. Youth mentors presented the study to potential participants in person at GRS offices in Alexandra and gave participants the opportunity to ask questions prior to completing the consent process. For the adolescent participants, interviews were conducted until the researchers felt that the data had conceptual depth in response to the research objectives [49–51]; they made the decision to stop data collection after 12 interviews. For key informant interviews, all eligible GRS youth mentors and staff, 12 in total, were invited and agreed to participate. This approach was appropriate given that all eligible male staff and mentors were actively involved in GRS programs and readily available.

## Data collection

Semi-structured interviews were conducted in participants' preferred languages, primarily English, Sesotho, and isiZulu, between March 1 and 30, 2024. The interviews were carried out by two researchers: 1) a former GRS youth mentor and an Alexandra resident with training in qualitative methods, who conducted interviews with adolescent boys and youth mentors; and 2) GRS's mental health advisor, who has a background in public health and education and training in qualitative research, conducted interviews with staff. The adolescent participants had no prior relationship with the researchers, whereas the GRS youth mentors and staff were familiar with them through their organizational roles. A lecturer from Stellenbosch University advised on the interview guides. The in-depth interview guide for adolescent participants was pretested and refined to improve clarity and cultural relevance before its use, and both versions are provided as Supporting Information (Appendix B – Interview Guide; C – Interview Guide Before Pretesting). The guide included open-ended questions and prompts exploring boys' everyday lives, challenges, and perspectives on mental health (e.g., "How would you describe mental health?"). It also covered background topics such as family, friends, school, and life in Alexandra, followed by questions about emotions, stress, coping strategies, and responses to difficult experiences. Additional questions focused on gender expectations, support systems, role models, and hopes for the future, concluding with boys' attitudes toward seeking help.

The key informant interview guide explored similar themes but focused more on informants' observations of adolescent boys, community norms, and local services, drawing on youth mentors' and staff members' experiences working with boys and their own shared background as slightly older male residents of Alexandra. Both interview guides were also designed to prompt personal reflections on these topics. Interviews took place in a quiet, private room at the GRS offices in Alexandra. Each participant took part in one individual, semi-structured interview. Interviews lasted 45–70 minutes and were audio-recorded.

## Data analyses

The interviews were transcribed verbatim, translated, and analyzed via reflexive thematic analysis [43,52]. Coding was inductive, with themes refined iteratively. Atlas.ti software (version 23.1.0) facilitated data organization. For interviews conducted in English (n = 12), Otter.ai supported transcription, but all transcripts were reviewed for accuracy by the research team. The interviews conducted in local languages were manually transcribed. Two researchers collaborated throughout the analysis to enrich interpretation [43] by coding a subset of transcripts for discussion and reviewing and discussing the coding scheme as it developed. While no major disagreements arose, the two researchers discussed their different perspectives throughout the coding process to enrich interpretation and highlight what felt most important in the data. In total, 92 codes and 13 code categories or preliminary themes were created in Atlas.ti. Preliminary themes were presented to the GRS youth mentors and staff during a half-day workshop as a form of member checking [53], with refinements made thereafter.

To enhance the trustworthiness of the findings, several strategies aligned with Guba and Lincoln's framework [54] were employed. Credibility was strengthened by triangulating data from multiple participant groups (adolescent boys, youth mentors, and male staff) and conducting member checking with staff and coaches. Transferability was addressed by providing a detailed account of the study's context, participants, and procedures. Dependability was supported through thorough documentation of all the data collection and analysis steps. Finally, confirmability was supported by maintaining an "*audit trail*" that included interview transcripts, coding notes, and reflexive notes to try and ground interpretations and conclusions in participants' accounts.

## Researcher positionality and reflexivity

Throughout the research process, the research team recognized how their identities and experiences shaped the data collection, analysis, and interpretation process. The lead researcher, a white American male in his 40s and a senior GRS staff member, interviewed staff. His senior position created the potential for power imbalances, especially when speaking with younger staff. In contrast, the research assistant, a Black South African male from Alexandra and a former GRS educator, interviewed adolescent boys and youth mentors. His shared background helped build rapport and trust, and create a safe, open space for discussion, but sometimes made him hesitant to probe sensitive issues for fear of causing distress.

Reflexive thematic analysis was conducted collaboratively, not to achieve consensus but to enrich interpretation, moving between semantic (surface meaning) and latent (underlying meaning) coding [43]. Throughout the analysis, we kept reflection notes on our assumptions and decisions, including efforts to make abstract concepts concrete in interviews by using familiar language and relatable scenarios. This active, interpretive stance underscores that the findings are not a neutral "discovery" of meaning, but co-constructed by participants and researchers within their unique life contexts, and through the analytic process [55].

## Ethical considerations

Ethical approval was granted by the Human Research Ethics Committee at Stellenbosch University (HREC Reference No: S23/08/197). The participants were informed of confidentiality measures, voluntary participation, and their right to withdraw at any time. A referral protocol for psychosocial support was in place, although no adverse events were reported,

one participant later asked to be accompanied to see a social worker. Informed consent was obtained from all participants, with parental or guardian permission obtained from participants under 18 years of age. Consent forms were available in English, isiZulu, and Sesotho.

## Results

Results are presented across the study's three aims, each with 2–5 themes that reflect ABYM's understanding, lived experiences, and strategies for navigating mental health in Alexandra. The aims and themes are summarized in Table 1 and detailed below. Pseudonyms are used for the attribution of direct quotes.

### Perceptions of mental health and the life experiences that shape it

The participants held diverse and varied beliefs about mental health, expressed uncertainty about the meaning of mental health and related topics, and blended social, psychological, and supernatural conceptualizations of mental health.

**Uncertainty about the meaning of mental health.** Many adolescent participants reported limited or no exposure to the term 'mental health,' which held little meaning to them unless they were explicitly exposed to it, usually through school programs, social workers, or community initiatives. As *Thato* (a 15-year-old male living in Alexandra) said, "*I've heard about it [mental health, but I don't know what it's about.*" The participants who understood the term 'mental health' often associated it with mental illness or negative attributes. The concept of mental illness was confusing to participants, who struggled to identify the line between emotional difficulties and a diagnosable mental illness. Some participants further described confusion about the difference between the difficulties of life in Alexandra and mental health. Nkosi (a 33-year-old male GRS staff member who lives and works in Alexandra) described this confusion as follows: *"The concept of mental health is still a foreign language for most people here in Alex. Not only for young people. We're all still trying to navigate the difference between mental health and our normal way of living."*

**Social understanding of mental health.** Most participants' mental health perceptions were tied to tangibles, lived experiences and their social environment. Xolani (a 28-year-old male GRS staff member living and working in Alexandra) described how some boys perceived Alexandra, "*they dream of getting out of this hood. But the thing is, they can't make ends meet, so they can escape.*" As Mfundo (a 40-year-old male GRS staff member who grew up in Alexandra) said, "*Adolescent boys do not sit back, think about life and appraise their overall psychological state. Their conceptualization of mental health is tied to concrete experiences and expressed in their own way.*" Mfundo continued by mentioning social factors when asked about adolescent boys' common mental health challenges, "*Mental health is about all these issues around where they live…the impact of poverty on their mental health, the impact of their experiences of violence.*"

**Table 1. Aims and Themes: ABYM's mental health perceptions and experiences.**

| | Aim | Themes |
|---|---|---|
| 1 | Perceptions of mental health and the life experiences that shape it | • Uncertainty about the Meaning of Mental Health<br>• Social, Psychological, Supernatural, and Racial Conceptualizations of Mental Health. |
| 2 | Masculine norms and personal beliefs, and their influence on perceptions of mental health | • Conflicts Between Masculine Norms, Personal Beliefs, and Lived Experiences.<br>• Discouraging Vulnerability and Mental Health.<br>• Conceptualization and Pressures of Being a Provider. |
| 3 | Coping strategies used in response to mental health challenges and beliefs that enable or limit help-seeking | • Self-care and Positive Coping Skills.<br>• Using Drugs and Alcohol as Coping Mechanisms.<br>• Help-Seeking Barriers: Mental Health Stigma, Problem Identification, Awareness, and Accessibility, and Supernatural Problems Require Supernatural Solutions. |

**Psychological terminology used in a nonclinical sense.** Despite limited exposure to the term 'mental health,' the participants tended to be more familiar with terms such as stress, depression, and anxiety. However, they were employed in everyday contexts rather than in clinical ones. As Kagiso (an 18-year-old male living in Alexandra) said, "*I get depressed when I have to fetch my school report. What if I failed*?" Mental health challenges were often seen as linked to or even caused by "*overthinking.*" Some participants suggested avoiding thinking or occupying the mind as ways to address mental health challenges. Sipho (a 15-year-old male living in Alexandra) said, "*People think a lot, and that damages your mind because the more you think, the more your problems appear*". Thabo (a 17-year-old male living in Alexandra), mentioned, "*I ended up getting sick from the stress. I developed neck pains from overthinking.*"

**Co-existence of supernatural understanding of mental health.** Many participants also attributed mental health challenges to supernatural causes, such as witchcraft or a spiritual calling. These beliefs did not contradict but coexisted with psychological or social explanations and perspectives on mental health. Lesedi (a 22-year-old GRS youth mentor living and working in Alexandra) described, "*In Alexandra, people believe in witchcraft, and sometimes that is not necessarily the case. Sometimes, some of the things that we go through are the things that we created for ourselves.*" However, Andile (a 31-year-old male GRS youth mentor living and working in Alexandra) highlighted how supernatural beliefs can obscure the actual causes of mental health challenges: "*They say that someone has been bewitched, or voodoo was used to make that person depressed, not knowing the full story or the real cause.*"

**Racial identity and mental health.** The participants' beliefs about cultural and racial identity impacted their mental health perceptions. Many participants perceived that "*Black communities*" often disregarded mental health, associating mental health with being "*for White people.*" Mandla (a 25-year-old male GRS youth mentor living and working in Alexandra) said, "*The Black Society says it is for White people, but why are we struggling with our mental health?*" Mfundo (a 40-year-old male GRS staff member) added that race is also linked to culture and history, which are overlooked factors in conceptualizations of mental health.

## Masculine norms and personal beliefs, and their influence on perceptions of mental health

The participants shared diverse perceptions of masculine norms, but it was clear that they had a profound influence on their self-esteem and identity, aspirations and goals, beliefs and behaviors, with many direct and indirect influences on their mental health. Thamsanqa (a 29-year-old GRS staff member) said, "*We're still figuring out what it means to be not healthy mentally versus what it means to be a man when you're not feeling okay.*" Nkosi (a 33-year-old male GRS staff member) talked about the idea of mental health being nonexistent for adolescent boys because of perceptions of masculinity and toughness. "*When you're a boy and you face challenges, you just have to overcome them. That's it. That is the nature of being a boy. So, mental health does not exist for them.*"

**Conflicts between norms and personal beliefs.** Many participants expressed internal conflicts and frustrations with masculine norms and rigid societal expectations that dictated their behavior as boys and men. These expectations shaped how participants perceived and engaged in recreational activities related to others and acceptable forms of expression. These expectations were often seen as preventing men and boys from being themselves. Xolani (a 36-year-old male GRS staff member living and working in Alexandra) said, "*Being a man growing up in Alex is like being put in a corner, in a box, to say, behave this way, or behave that way, for us to accept you.*" He added, "*There are boys who are sensitive, and those boys are afraid to show who they are.*"

**Discouraging vulnerability and mental health.** Masculine norms were perceived as discouraging vulnerability and promoting self-reliance. These expectations caused internalized pressure to appear strong and unemotional, which some described as a significant source of mental health challenges among males. Mandla (a 25-year-old male GRS youth mentor) connected masculine norms directly with mental health problems, saying, "*It is what makes mental health problems so common in the boys, because they say you must be the man, you must not show emotions, even when you are dying inside.*" Several participants also expressed clear costs for showing emotions to others as boys, with people

equating emotions with weakness. Bongani (an 18-year-old male living in Alexandra) said, "*People will say negative things [if I show emotions]. They might say I'm a sissy*".

Mfundo (a 40-year-old male GRS staff member) described internalized ideas of self-reliance and toughness as destructive and harmful:

> If you're not feeling great, the answer that most men provide is "*man up.*" It goes down to how you deal with yourself. If the only tool you have for the lowest moments in your life is to "*man up,*" you're going to abuse yourself to get through things, and this leads to much self-harm for men.

Vulnerability could even be discouraged in the most difficult circumstances, as one participant said, *"when my mom died, I just thought that there won't be anything good coming anymore. She was the pillar of everything…my brother would tell me to go in the room and get under the covers when I was crying."*

However, many participants challenged norms of self-reliance or stoicism, distancing their personal views from what they perceived as dominant masculine norms, describing these norms as "old," "traditional," or "cultural" with a negative connotation. Thato (a 15-year-old male living in Alexandra) described, "*I don't believe that you should act like a man or that boys don't cry because everyone cries when they feel pain.*"

**Conceptualization and pressures of being a provider.** Being economically successful and providing financial support in relationships were central to what participants believed was expected of them as boys. The expectation to be financially successful and provide for others was already present at a young age, with school perceived as the key pathway to success. Many participants reported intense pressure to do the right things now to provide for their families later. Families often remind boys that they need to succeed in school and life to lift them out of poverty. The participants described needing to "*finish school so I can support my family,*" "*buy them a house,*" or "*take care of them when I get older.*" Siyabonga (a 15-year-old male living in Alexandra) said of his family members, "*They tell me to take them out of the situation they are in. They want me to take them out of the life they are living and give them a better life.*"

Several participants also described significant pressure to fulfill the expectation of providing for their romantic interests. This expectation often led to stress, as they believed that failing to provide would result in losing their partners. Mbu (a 16-year-old male living in Alexandra) recounted being dumped by his girlfriend when he had no money, illustrating the painful impact of this pressure and the belief that "*no money, no love.*" The provider role in traditional heteronormative romantic relationships not only causes immense pressure for the male but also highlights that this expectation establishes harmful gender norms and roles at a young age. As Mfundo (40-year-old male GRS staff member) said:

> If you're 15 and you're socialized that as a man, I must provide, and you keep hearing that, now when you have a girlfriend, everyone tells you if you don't provide for your girlfriend, whatever that means when you're 15, then someone else will steal her. This is extremely problematic because it assumes that she does not have a mind and will of her own. It assumes a lot of things. However, that is what you're told, and what you live with. Then, you have to 'man up' and find a way to provide, which is an incredible amount of pressure to put on anyone, let alone a 15-year-old. It leaves you open to a lot of negative ideas and actions. You have to find a way to keep her. She's kept. That's the masculine idea.

### Coping strategies used in response to mental health challenges and beliefs that enable or limit help-seeking

The participants navigated mental health challenges through a mix of healthy and harmful coping mechanisms and identified drugs and alcohol as typical coping mechanisms in the community. Despite some acknowledgment of the societal expectations of self-reliance, many participants expressed comfort in talking about their feelings with trusted adults or peers.

**Self-care and positive coping skills.** On the positive side, participants identified recreational activities such as "*playing soccer*," "*listening to music*," "*drawing*," and "*martial arts*" as ways to "*relieve stress*," "*feel better*," or "*forget about everything else.*" These activities provided an outlet for their emotions and helped them manage their mental health more effectively. Participation in sports, in particular, was a popular way to relax and destress.

Despite the perceived masculine norm of self-sufficiency, many participants recognized the importance of talking to someone when facing difficulties. The participants identified various people they could talk to (e.g., "*mom*," "*dad*," "*friends*," "*sibling*") and described the importance of talking to a trusted and nonjudgmental person. Sibusiso (a 16-year-old male living in Alexandra) said, "*I talk to someone I can trust—someone who will keep whatever I tell him between us and won't judge me.*" Thato (a 15-year-old male living in Alexandra) mentioned, "*My grandmother is my therapist.*"

**Using drugs and alcohol as coping mechanisms.** Several participants identified drugs and alcohol as common coping mechanisms for people, especially men, in their community. Themba (a 15-year-old male living in Alexandra) acknowledged that smoking offers temporary relief: "*It benefits some because when they smoke or drink, they calm down.*" However, overall, participants described drugs and alcohol as harmful ways to cope with stress that can make problems worse. Xolani (a 36-year-old male GRS staff member) said, "*They've instilled in us to say, 'If you take this, you forget everything.' However, they forget that tomorrow, that same pain that you felt is probably ten times more.*" Some participants described drug and alcohol consumption as a signal of despair and hopelessness. Njabulo (a 23-year-old male GRS youth mentor living and working in Alexandra) said this about his male peers: "*They can't make their dreams come true. So, what they do is start smoking drugs, they start engaging in crime, they start drinking alcohol.*" Lesedi (a 22-year-old GRS youth mentor) reached a point where his discipline wasn't paying off, so why not drink and smoke? "*Why am I still stuck playing Ultra (local soccer league)? Maybe drinking water and eating healthy [doing the right thing] doesn't pay off. Let me drink and smoke.*"

**Help-seeking barriers: Mental health stigma.** The stigma surrounding mental health was described as a significant barrier to help-seeking. Many participants feared being judged or labeled weak if they admitted to needing help. The belief that mental health challenges are not a concern for Black people or that supernatural forces cause them was seen as discouraging them from seeking professional support. Nkosi (a 33-year-old male GRS staff member) also mentioned associating the local mental health clinic as "for the crazy ones", and Mandla (a 25-year-old male GRS youth mentor) said, "*Most of the problem for men is stigma. You do not want to seek help because you're thinking, eish, what will the people say?*"

**Help-seeking barriers: Problem identification and accessibility.** Another considerable barrier to help-seeking among participants was knowing when something was wrong, where to go for help, or what mental health support entails. Nkosi (a 33-year-old male GRS staff member) highlighted the need for more information on how to identify when someone needs help:

> I think the more we gain knowledge or information about mental health, the easier it will be to support someone and maybe try to reach them more easily. However, because we lack knowledge, we tend to turn a blind eye. We do not know how to spot when something's wrong with someone. That is why we find people committing suicide.

There was also a lack of awareness or interest in mental health services, with some participants unable to identify a single place to visit if they needed help. Thamsanqa (a 29-year-old GRS staff member living and working in Alexandra) highlighted that boys were not interested in the available services: "*They were not interested in coming to sit down and listen to someone talking about mental health because of how mental health is talked about… Therefore, it has made it difficult for boys to access mental health-related information.*" Luyanda (a 26-year-old male GRS youth mentor living and working in Alexandra) acknowledged a need for more mental health services in Alexandra: "*We need our own psychiatrists and social workers. We need them in Alex. We should have free therapists. If we did, men and boys would go to counseling.*"

**Help-seeking barriers: Supernatural problems and treatments.** Even though many participants held supernatural beliefs about mental health, they identified an overreliance on them as a significant barrier to accessing mental health support and services. Nkosi (a 33-year-old male GRS staff member) talked about his neighbor, a local soccer player, "*Right now, she's not okay mentally. And, of course, they say it is witchcraft, and they're not thinking of mental health and possibly trying to get this young girl to a mental institution.*" Karabo (a 26-year-old male GRS youth mentor living and working in Alexandra) stated bluntly, "*If it's supernatural, you will go in the wrong direction and seek help in the wrong places. If you say it is witchcraft, that means you go to traditional healers, and they do not know anything about mental health.*"

## Discussion

This study highlights that ABYM in Alexandra often expressed uncertainty about the concept of mental health, largely due to unfamiliarity with the term itself. While GRS staff and youth mentors demonstrated more knowledge, they noted that many boys had never encountered the term because mental health is often perceived locally as "irrelevant for males" or "for White people." Such perceptions contributed to broader confusion about distinguishing mental health challenges from what is considered "normal life" for males growing up in Alexandra. The prevailing attitude that men and boys must "just deal with" challenges reinforced the invisibility of their mental health needs.

Participants' understanding of mental health was mostly grounded in everyday language, using terms like "pressure," "stress," and "overthinking" to describe mental health in a general sense. This aligns with global research indicating that adolescents often describe psychological distress using nonclinical language [37,38]. This preference for familiar, concrete expressions also guided the research team's interviewing approach, which used prompts in local languages about emotions or feelings, or in the case of masculinity, "what does it mean when someone says act like a man" to elicit reflections on masculinity and mental health. While some participants were familiar with the term mental health, they frequently associated it with mental illness. This mirrors global findings where adolescents equate mental health with severe mental disorders and see it as disconnected from daily life [37].

Socio-cultural factors, such as poverty, violence, and spiritual beliefs, also shaped how participants understood mental health and masculinity. Cultural beliefs about witchcraft or punishment for ignoring spiritual callings co-existed with psychological and social understandings, underscoring the need for local frameworks for mental health that integrate local explanatory models with psychological concepts [56]. Masculinity was a central social factor and lens through which participants interpreted mental health. Masculine norms promoting emotional suppression and self-reliance were key barriers to emotional expression and help-seeking, echoing global literature on masculinity and men's health [15,57]. Masculinity and mental health also influence each other bidirectionally. For example, psychological distress may influence how young men evaluate their masculinity, leading to more negative self-assessments [58]. This was evident in several participants' comments about the self-abuse that occurs when young men tell themselves to "man-up" in their lowest moments. Unlike much of the research in South Africa that has examined masculinity in relation to power, gender, and harmful behaviors [26,59], this study contributes new insights into how ABYM describe the emotional costs of conforming to narrow gender expectations, such as the pressures of male provision and the internal tensions they experience.

When asked about masculinity, participants often discussed it in terms of identity and self-expression rather than in the context of gender relationships. Many participants expressed that they felt "put in a corner, in a box" by societal rules dictating acceptable behavior and interests. These expectations extended beyond broad virtues like toughness to everyday activities, where, for example, playing soccer was "masculine" but watching anime or drawing was not. While some participants upheld traditional ideals like self-reliance and toughness, many rejected these as "old" or "traditional", instead sharing views that encouraged "sharing your feelings", "asking for help", and "showing emotions because we're all human." As masculine norms change, so do their meaning and health implications for boys [13], and the participants' voiced beliefs that possibly signal shifts in attitudes and a desire for redefining masculinity, something found in other studies [13,19].

Boys described a range of coping strategies, from sports, music, and drawing to substance use. Positive male coping strategies exist, provide relief, and should be leveraged, aligning with evidence that reinforcing coping skills can enhance male engagement with mental health support [30]. Substance use, however, was viewed as both a common and harmful way to manage stress, often learned from older male relatives and normalized within the community.

Supportive relationships with peers, mentors, and family members were recognized as important. Yet, boys experienced tension between personal beliefs favoring help-seeking and masculine norms promoting self-reliance. This conflict was amplified by the limited availability of youth- and male-friendly services, common barriers echoed in several studies on youth help-seeking [60,61]. There is also a shortage of Black mental health professionals, particularly those who can communicate in the many local languages [18]. Even when services existed, they were often perceived as irrelevant or stigmatized. Stigma operated through intersecting beliefs: age hierarchy ("what do you know about loneliness as a young person?"), race ("counselling is for white people"), gender ("as a boy they'll call you a sissy"), and mental illness stereotypes ("crazy," "mad"). Such nuanced stigma requires further attention and creates unique and layered barriers to care, reinforcing the need for tailored messaging and outreach to boys, especially safe spaces and peer networks [30,62].

## Limitations

This study had several limitations. The main limitation was the lack of generalizability of the findings, as they may be specific to the sample of participants interviewed in Alexandra, though the depth provided by our in-depth exploration is also a study strength [63]. The sample size could have limited the diversity of perspectives and experiences captured, potentially overlooking important variations within the population. The purposive sampling of adolescent boys ensured that participants had relevant experiences with the topic, but it may have excluded the perspectives of boys who were less engaged with GRS programming. Social desirability bias is possible, where participants may have given responses they thought the interviewers wanted to hear. Another limitation was translating interviews from Sesotho and isiZulu into English. It is often difficult to fully express emotions and experiences in one's mother tongue, and an additional layer of translation may have altered participants' intended meanings. These limitations were balanced by efforts to enhance trustworthiness through triangulation across participant groups, member checking with youth mentors, and maintaining an audit trail of analytic decisions.

## Recommendations for intervention design

Based on findings from this study, we propose the following recommendations for developing mental health interventions for ABYM in Alexandra, South Africa.

1. **Co-design interventions with ABYM**: Engage boys as collaborators in developing mental health content, messages, and implementation. Our findings demonstrate the critical importance of ABYM's voices in contributing to developing solutions for the mental health challenges they face. Co-designing culturally relevant solutions ensures that interventions resonate with the lived experiences and preferences of ABYM, leading to greater engagement and impact [60,64]. Co-designing interventions could include youth advisory panels, co-design workshops, or participatory media projects that provide a platform for ABYM to develop campaigns that align with their language and beliefs.

2. **Integrate mental health into sports and recreation programs.** Embed mental health support within activities and spaces that boys already attend. Sports-based interventions have been successful in creating informal support networks where boys can naturally discuss personal issues [62] and can normalize mental health conversations, allowing boys to access support where they are. This could include providing basic training in mental health literacy for sports coaches to facilitate mental health check-ins before and after practices and games.

3. **Build the capacity of positive male mentors and role models.** Equip male mentors, coaches, and community leaders to model empathy, care, and openness as foundational aspects of healthier masculinity. When respected men share their own mental health stories, it challenges stigma and reinforces the acceptability of seeking help. This could include intergenerational dialogues about masculinity and mental health and mental health training for male community leaders.

4. **Implement peer campaigns that promote healthy coping and emotional openness.** Awareness campaigns targeting ABYM should highlight the positive aspects of mental health, such as self-awareness, resilience, and emotional regulation, reframing mental health as something everyone has and can strengthen [65–67]. These campaigns should also address the unique and intersecting forms of stigma that boys and young men experience, including racialized beliefs that mental health is "for White people" and fears of being perceived as weak. Building on healthy self-care and coping strategies already used by boys, interventions can incorporate practical skills such as problem-solving, relaxation techniques, stress management, and emotional expression, delivered in culturally relevant and accessible formats. In contexts where professional services are limited, peer-led initiatives that model these skills and normalize emotional openness can offer a practical first step toward improved mental health outcomes [68,69].

5. **Create locally relevant and youth-friendly referral pathways**. Collaborate with schools, social workers, health providers, and community organizations to develop referral systems that are accessible, trusted, and tailored to boys. In addition to in-person options, offer digital and private channels, such as confidential WhatsApp lines, SMS-based helplines, or mobile platforms, that boys can use anonymously to seek information or connect with mental health professionals. These approaches can help overcome stigma, reduce fear of judgment, and make help-seeking easier [70].

6. **Develop interventions that promote access to cultural and historic knowledge**. Mental health interventions and messaging should recognize the role of cultural and historical knowledge in fostering identity, belonging, and meaning among ABYM in Alexandra. Research indicates that positive meaning and feelings toward one's racial identity are protective for psychological well-being among Black youth, with associations with higher self-esteem, better emotional adjustment, and reduced internalizing and externalizing symptoms [71–73]. Given that much of this evidence is drawn from non–South African contexts, there is a clear need to develop and evaluate locally grounded interventions that strengthen cultural and historical knowledge as a pathway to improved mental health outcomes among ABYM in South Africa.

## Conclusion

This study examined how ABYM in Alexandra understand and experience mental health, using everyday language shaped by their social environment and masculine norms. While some internalized traditional expectations of self-reliance, others expressed a desire to redefine masculinity to include emotional openness, vulnerability, and help-seeking. These findings point to the need for interventions that reflect boys' lived realities, address the social determinants of mental health, and move beyond solely clinical approaches. Although based on a small sample, the study revealed a nuanced mix of views that both question and uphold dominant masculine norms, as well as cultural or spiritual conceptualizations of mental health. This underscores the importance of developing locally grounded language, concepts, and approaches to mental health.

## Supporting information

**S1 File. COREQ (Consolidated Criteria for Reporting Qualitative Research) checklist completed for this study.**
(DOCX)

**S2 File. Draft semi-structured interview guide used for pre-testing with adolescent boys and young men.**
(DOCX)

**S3 File. Final semi-structured interview guide used for in-depth interviews with adolescent boys and young men.**
(DOCX)

## Acknowledgments

We gratefully acknowledge the young people, coaches, staff, communities, and partners who make Grassroot Soccer's mental health programs possible. Their commitment, insight, and everyday efforts are the foundation of this work. We also acknowledge the contributions of the Center for Dissemination and Implementation Science at the University of Chicago's Department of Medicine.

## Author contributions

**Conceptualization:** Christopher Barkley.

**Data curation:** Christopher Barkley, Sandile Mnculwane.

**Formal analysis:** Christopher Barkley.

**Investigation:** Sandile Mnculwane.

**Methodology:** Christopher Barkley, Katherine G Merrill.

**Supervision:** Zuhayr Kafaar.

**Writing – original draft:** Christopher Barkley.

**Writing – review & editing:** Katherine G Merrill, Zuhayr Kafaar.

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
