## [Decision Letter · Decision Letter 0]

9 Jul 2025

PMEN-D-25-00230

“Being a man is like being put in a box”: A qualitative study of adolescent boys’ understanding and experiences of mental health in an urban community in South Africa

PLOS Mental Health

Dear Dr. Barkley,

Thank you for submitting your manuscript to PLOS Mental Health. After careful consideration, we feel that it has merit but does not fully meet PLOS Mental Health’s publication criteria as it currently stands. Therefore, we invite you to submit a revised version of the manuscript that addresses the points raised during the review process.

Please note that we have only been able to secure a single reviewer to assess your manuscript. We are issuing a decision on your manuscript at this point to prevent further delays in the evaluation of your manuscript. Please be aware that the editor who handles your revised manuscript might find it necessary to invite additional reviewers to assess this work once the revised manuscript is submitted. However, we will aim to proceed on the basis of this single review if possible.

Please attend to the reviewer's concerns regarding greater methodological detail, as well as substantial expansion of the discussion section, including clearly stated limitations.

We look forward to receiving your revised manuscript.

Kind regards,

Avanti Dey, PhD

Staff Editor

PLOS Mental Health

Journal Requirements:

Additional Editor Comments (if provided):

Reviewers' comments:

Reviewer's Responses to Questions

**Comments to the Author**

1. Does this manuscript meet PLOS Mental Health’s publication criteria?

Reviewer #1: Yes

2. Has the statistical analysis been performed appropriately and rigorously?

Reviewer #1: Yes

3. Have the authors made all data underlying the findings in their manuscript fully available (please refer to the Data Availability Statement at the start of the manuscript PDF file)?

Reviewer #1: Yes

4. Is the manuscript presented in an intelligible fashion and written in standard English?

Reviewer #1: Yes

Reviewer #1: As a reviewer, I have identified several areas that require attention to enhance the quality of the manuscript. My suggestions are as follows:

1. The manuscript lacks a conclusive section and actionable recommendations. Adding these would strengthen the paper's impact (line no. 47).

2. The use of purposive sampling is noted, but justification for its selection over convenience sampling or data saturation is required. Please clarify the rationale behind this choice (line no. 146).

3. The results section should follow the order of objectives outlined in the introduction for better coherence and flow (line no. 209).

4. The discussion section is underdeveloped. Each result should be thoroughly discussed in this section to provide depth and context (line no. 430).

5. The discussion lacks reflective insights. Incorporating reflections would add value to the manuscript (line no. 430).

6. The manuscript does not adequately address the limitations of the study. A detailed discussion on limitations is essential for transparency and credibility (line no. 502).

7. The references cited are outdated. It is recommended that at least 90% of the references be from the period between 2015 and 2024/2025 to reflect current research trends and findings (line no 521).

**Do you want your identity to be public for this peer review?** For information about this choice, including consent withdrawal, please see our Privacy Policy

Reviewer #1: No

---

## [Decision Letter · Decision Letter 1]

29 Sep 2025

PMEN-D-25-00230R1

“Being a man is like being put in a box”: A qualitative study of adolescent boys’ and young men's understanding and experiences of mental health in an urban community in South Africa

PLOS Mental Health

Dear Dr. Barkley,

Thank you for submitting your manuscript to PLOS Mental Health. After careful consideration, we feel that it has merit but does not fully meet PLOS Mental Health’s publication criteria as it currently stands. Therefore, we invite you to submit a revised version of the manuscript that addresses the points raised during the review process.

Your manuscript has been evaluated by one reviewer, and they request a completed COREQ checklist be included with the manuscript.

Please note that we have only been able to secure a single reviewer to assess your manuscript. We are issuing a decision on your manuscript at this point to prevent further delays in the evaluation of your manuscript. Please be aware that the editor who handles your revised manuscript might find it necessary to invite additional reviewers to assess this work once the revised manuscript is submitted. However, we will aim to proceed on the basis of this single review if possible.

We look forward to receiving your revised manuscript.

Kind regards,

Jenna Scaramanga

Staff Editor

PLOS Mental Health

Journal Requirements:

Additional Editor Comments (if provided):

Reviewers' comments:

Reviewer's Responses to Questions

**Comments to the Author**

Reviewer #1: All comments have been addressed

publication criteria?

Reviewer #1: Yes

3. Has the statistical analysis been performed appropriately and rigorously?

Reviewer #1: Yes

4. Have the authors made all data underlying the findings in their manuscript fully available (please refer to the Data Availability Statement at the start of the manuscript PDF file)?

Reviewer #1: (No Response)

5. Is the manuscript presented in an intelligible fashion and written in standard English?

Reviewer #1: Yes

Reviewer #1: 1. Need to do incorporate COREQ (Consolidated criteria for Reporting Qualitative research) Checklist or other standardized checklist.

**Do you want your identity to be public for this peer review?** For information about this choice, including consent withdrawal, please see our Privacy Policy

Reviewer #1: No

---

## [Decision Letter · Decision Letter 2]

6 Jan 2026

PMEN-D-25-00230R2

“Being a man is like being put in a box”: A qualitative study of adolescent boys’ and young men's understanding and experiences of mental health in an urban community in South Africa

PLOS Mental Health

Dear Dr. Barkley,

Thank you for submitting your manuscript to PLOS Mental Health and I hope 2026 has started well. I am very sorry for the delay, which was due to difficulties finding the second reviewer and my leave over the festive season.  We now have the second report and after careful consideration of it, we would like to invite you to submit a revised version of the manuscript that addresses the points raised during the review process. The points that are raised are minor and so the revision will be assessed by in-house, rather than sending this back out to review again.

Please ensure that you fully address all of the comments, which you can find at the end of this email.

We look forward to receiving your revised manuscript.

Kind regards,

Dr Karli Montague-Cardoso

Staff Editor

PLOS Mental Health

Journal Requirements:

Reviewers' comments:

Reviewer's Responses to Questions

**Comments to the Author**

Reviewer #1: All comments have been addressed

Reviewer #2: (No Response)

publication criteria?

Reviewer #1: Yes

Reviewer #2: Yes

3. Has the statistical analysis been performed appropriately and rigorously?

Reviewer #1: Yes

Reviewer #2: Yes

4. Have the authors made all data underlying the findings in their manuscript fully available (please refer to the Data Availability Statement at the start of the manuscript PDF file)?

Reviewer #1: Yes

Reviewer #2: Yes

5. Is the manuscript presented in an intelligible fashion and written in standard English?

Reviewer #1: Yes

Reviewer #2: Yes

Reviewer #1: Good work

Reviewer #2: This is a very engaging article that provides an important contribution to the literature on mental health. The author/s present an interesting analysis of how adolescent boys grapple with the challenges of mental health while negotiating their masculine identities. Drawing on South African and international literature the authors illuminate the complex ways in which adolescent boys and young men negotiated constructions of masculinity alongside the recognition of mental health challenges and the pursuit of support. The study explored the coping mechanisms that boys utilized in response to mental health challenges and offers suggestions for mental health interventions. The implications that the findings in this article have for adolescent boys’ mental health should make this article of much interest to this journal’s readership.

Suggestions for Revision

The introduction and literature reviewed work well and concisely provide the empirical and conceptual rationale for this study. My main suggestion for revision pertain to some of the analysis.

The paper would benefit from incorporating additional interview excerpts from the adolescent participants to more clearly substantiate the analytical claims made in the analysis and discussion. Greater use of participants’ own words would strengthen the depth of the arguments.

The authors discuss masculine norms as though they are fixed and uniform. However, constructions of masculinity are fluid, relational, and contextually produced. Social, cultural, and structural factors play a significant role in shaping how boys and men construct their masculine identities. Including an early section that outlines the nature of masculinity and identifies the forms of masculinity that are hegemonic within the Alexandra context would provide a stronger conceptual foundation for the arguments that follow.

In the analysis and discussion, the authors appear to suggest a unidirectional relationship in which masculinity shapes mental health, as though masculinity pre-exists mental health. However, mental health experiences often precede and actively inform the construction of masculine identities. It would therefore strengthen the paper for the authors to also consider how the boys’ mental health experiences shaped, mediated, and were implicated in the ways they understood and performed masculinity.

Some of the boys asserted that mental health issues are for White people. This points to important racialised understandings of mental health that warrant deeper analysis. Given South Africa’s historical and socio-political context, a more sustained discussion of the racial overtones shaping how mental health is perceived, accessed, and legitimised would significantly strengthen the paper.

Some specific points to consider revising:

Sampling and Recruitment

[1] Providing a brief background of the adolescent participants would add depth and contextual richness to the subsequent analysis, allowing readers to better situate the boys’ experiences and narratives.

[2] How many interviews were conducted with each of the applicants.

[3] In addition to clinical ethical considerations, the authors should clarify what other safeguards were put in place to protect participants from potential emotional or psychological harm, considering the sensitive and personal topics explored in the study.

[4] While the authors briefly note this complexity in the conclusion, the paper does not sufficiently engage with the nuanced realities of the Alexandra context, particularly the interplay of varied racial and cultural belief systems in the construction of masculinities and the intersection with mental health, especially, in the analysis and discussion.

**Do you want your identity to be public for this peer review?** For information about this choice, including consent withdrawal, please see our Privacy Policy

Reviewer #1: No

Reviewer #2: **Yes:** Vijay Hamlall

---

## [Editor Report · Decision Letter 3]

14 Jan 2026

“Being a man is like being put in a box”: A qualitative study of adolescent boys’ and young men's understanding and experiences of mental health in an urban community in South Africa

PMEN-D-25-00230R3

Dear Mr. Barkley,

We are pleased to inform you that your manuscript '“Being a man is like being put in a box”: A qualitative study of adolescent boys’ and young men's understanding and experiences of mental health in an urban community in South Africa' has been provisionally accepted for publication in PLOS Mental Health.

Best regards,

Karli Montague-Cardoso

Staff Editor

PLOS Mental Health